# Structural basis for the assembly of the mitotic motor Kinesin-5 into bipolar tetramers

**Jessica E Scholey[†], Stanley Nithianantham[†], Jonathan M Scholey, Jawdat Al-Bassam***

Department of Molecular and Cellular Biology, University of California, Davis, Davis, United States

**Abstract** Chromosome segregation during mitosis depends upon Kinesin-5 motors, which display a conserved, bipolar homotetrameric organization consisting of two motor dimers at opposite ends of a central rod. Kinesin-5 motors crosslink adjacent microtubules to drive or constrain their sliding apart, but the structural basis of their organization is unknown. In this study, we report the atomic structure of the bipolar assembly (BASS) domain that directs four Kinesin-5 subunits to form a bipolar minifilament. BASS is a novel 26-nm four-helix bundle, consisting of two anti-parallel coiled-coils at its center, stabilized by alternating hydrophobic and ionic four-helical interfaces, which based on mutagenesis experiments, are critical for tetramerization. Strikingly, N-terminal BASS helices bend as they emerge from the central bundle, swapping partner helices, to form dimeric parallel coiled-coils at both ends, which are offset by 90°. We propose that BASS is a mechanically stable, plectonemically-coiled junction, transmitting forces between Kinesin-5 motor dimers during microtubule sliding.

*For correspondence: jawdat@ucdavis.edu

†These authors contributed equally to this work

Competing interests: The authors declare that no competing interests exist.

## Introduction

Accurate chromosome segregation during mitosis is known to underlie the propagation of all cellular life. This process depends upon the action of a bipolar, macromolecular machine, the mitotic spindle, which uses dynamic microtubules (MTs) plus multiple kinesin and dynein motors to generate the piconewton-scale forces required for mitotic movements (*Loughlin et al., 2008*; *Goshima and Scholey, 2010*; *Walczak et al., 2010*; *McIntosh et al., 2012*). Of these, the MT-based motor, Kinesin-5 plays a key role, being essential for the assembly of bipolar spindles in most eukaryotic cells and driving or constraining the rate of spindle elongation (*Enos and Morris, 1990*; *Sawin et al., 1992*; *Saunders et al., 2007*; *Brust-Mascher et al., 2009*). Purified, native Kinesin-5 is a 'slow', plus-end-directed, bipolar, homotetrameric motor capable of using pairs of N-terminal motor domains at opposite ends of a central rod to crosslink MTs into bundles (*Cole et al., 1994*; *Kashina et al., 1996a, 1996b*; *Gordon and Roof, 1999*; *Acar et al., 2013*). Moreover, single Kinesin-5 holoenzymes can crosslink and slide adjacent MTs, displaying a threefold preference for MTs in the antiparallel vs the parallel orientation (*Hentrich and Surrey, 2010*; *Kapitein et al., 2005*; *van den Wildenberg et al., 2008*; *Weinger et al., 2011*). Together, these data support the hypothesis that Kinesin-5 functions via a 'sliding filament' mechanism, crosslinking adjacent MTs into bundles throughout the spindle (*Sharp et al., 1999*), and exerting outward or braking forces on antiparallel MTs to coordinate bipolar spindle assembly and to drive or constrain poleward flux and anaphase B spindle elongation (*Valentine et al., 2006a*; *Kaseda et al., 2009*; *Scholey, 2009*; *Subramanian and Kapoor, 2012*). In this model, the assembly of four Kinesin-5 polypeptides into a bipolar, tetrameric minifilament is crucial, but the molecular basis of this arrangement, which is, so far as we know, unique among MT-based motor proteins, remains unclear.

The kinesin superfamily of MT-based motors is organized into 14 or more families (*Vale, 2003*; *Lawrence et al., 2004*; *Wickstead and Gull, 2006*; *Hirokawa et al., 2009*). A characteristic feature of

**eLife digest** Successful cell division requires copies of the chromosomes containing the genetic material of a cell to be accurately copied and then separated so that when a cell divides, each new daughter cell contains exactly one copy of each chromosome. If this does not happen, the cell may malfunction or die.

To separate the duplicated chromosomes, a biological machine called the mitotic spindle forms inside the cell. This has two poles, one at each end, with each pole being responsible for gathering together the chromosomes for delivery to each of the daughter cells. Large numbers of long, thin protein tubes called microtubules extend out of each pole. Some microtubules attach to the chromosomes, whilst others are responsible for pushing apart the two poles—and the chromosomes attached to them—to the opposite sides of the cell before it divides.

To move the poles, motor proteins slide pairs of microtubules that are attached to opposite poles over each other. The Kinesin-5 family of motor proteins is particularly important for mitosis, because it is essential for forming the mitotic spindle and for making it work correctly. These motors assemble into motile machines that can apply a force to both of the microtubules in a sliding pair at the same time because they contain motor units at each end connected by a central rod.

The structure of this central rod is crucial for the successful operation of Kinesin-5. Scholey, Nithianantham et al. have now worked out the structure of a region of this filament called the bipolar assembly, or BASS domain. This structure is more complicated than expected: it contains four helixes made of protein that are all intertwined with each other.

In addition, Scholey, Nithianantham et al. found two 'molecular pockets' that small molecules can access. By entering the pockets, the molecules could disrupt the structure of the BASS domain, and consequently prevent Kinesin-5 from forming the dual-ended machines required to work properly. As Kinesin-5 is required to build the mitotic spindle, this would interfere with cell division. Targeting molecules into these pockets could therefore potentially form part of an anti-cancer therapy, preventing the rapid cell divisions behind the spread of the disease.

kinesin motors is the presence of a pair of adjacent motor domains capable of hydrolyzing ATP to walk in a hand-over-hand fashion along a MT track, thereby generating force and motion (*Vale and Milligan, 2000*). The adjacent motor domains are oriented by a parallel, α-helical coiled-coil rod that is capable of transmitting forces generated by the pairs of motor domains at one end of the rod to the associated cargo at the other end. The striking bipolar organization of Kinesin-5 family motor proteins means that the cargo for each motile unit (i.e., the pair of motor domains stepping along a MT protofilament) is another motile unit located at the opposite end of the bipolar minifilament. Therefore, the central rod must be able to transmit forces between pairs of motor domains located at its opposite ends in order to drive or constrain MT–MT crosslinking and sliding within the mitotic spindle, but how this is accomplished remains unclear. Moreover, because the homotetrameric architecture of Kinesin-5 appears to be required for MT–MT crosslinking and for normal mitosis and cell division (*Hildebrandt et al., 2006*; *Tao et al., 2006*), it is plausible to think that the disruption of its bipolar, oligomeric state could inhibit the rapid divisions of proliferating cancer cells, thereby providing a strategy for cancer therapy (*Owens, 2013*). Therefore, understanding how Kinesin-5 adopts its bipolar architecture is an important problem whose solution could lead to medical applications in addition to advancing our understanding of the basic mechanisms of motor protein function and mitosis.

Structure-function analysis of the *Drosophila melanogaster* Kinesin-5, KLP61F, identified a central bipolar assembly (BASS) domain within the coiled-coil rod that is critical for this motor's bipolar organization and whose deletion results in disassembled, mostly monomeric, Kinesin-5 (*Acar et al., 2013*). It is likely that the ability of this domain to organize four Kinesin-5 subunits into a bipolar minifilament and to transfer forces between pairs of motor domains situated at opposite ends of a 60-nm rod is essential for Kinesin-5 activity and mitotic cell viability (*Hildebrandt et al., 2006*; *Tao et al., 2006*). To better understand how the BASS domain promotes the assembly of Kinesin-5 into bipolar tetramers, we determined its structure using X-ray crystallography and validated the structure using structure-based mutational analysis. Instead of the expected antiparallel arrangement of two side-by-side parallel coiled-coils proposed previously (*Kashina et al., 1996a*; *Acar et al., 2013*), we find that the BASS

domain has a novel four-helix bundle structure, which very likely has important implications for its mechanism of action within the spindle.

## Results and discussion

### Overall structure of the Kinesin-5 BASS domain

We purified a minimal BASS domain construct that assembles into a tetrameric, 220 Å long rod based on EM (KLP61F residues 635–835; *Figure 1*) (*Acar et al., 2013*). This BASS tetramer formed hexagonal (P6₄22) crystals, which were used to determine its structure to 2.6-Å resolution using X-ray crystallography with the single wavelength anomalous dispersion (SAD) method from selenomethionine-and mercury-derived crystals (Materials and methods; *Table 1*). The asymmetric unit contains a canonical anti-parallel coiled-coil BASS domain dimer encompassing residues 640–802 of both chains, which represents the basic assembly unit of a Kinesin-5 minifilament (*Figure 1*, *Figure 1—figure supplement 1*). This anti-parallel dimer is associated with a second dimer through a crystallographic dyad axis to form the full tetramer, whose dimer–dimer packing surface extends over a large area (≈7000 Å²) and is

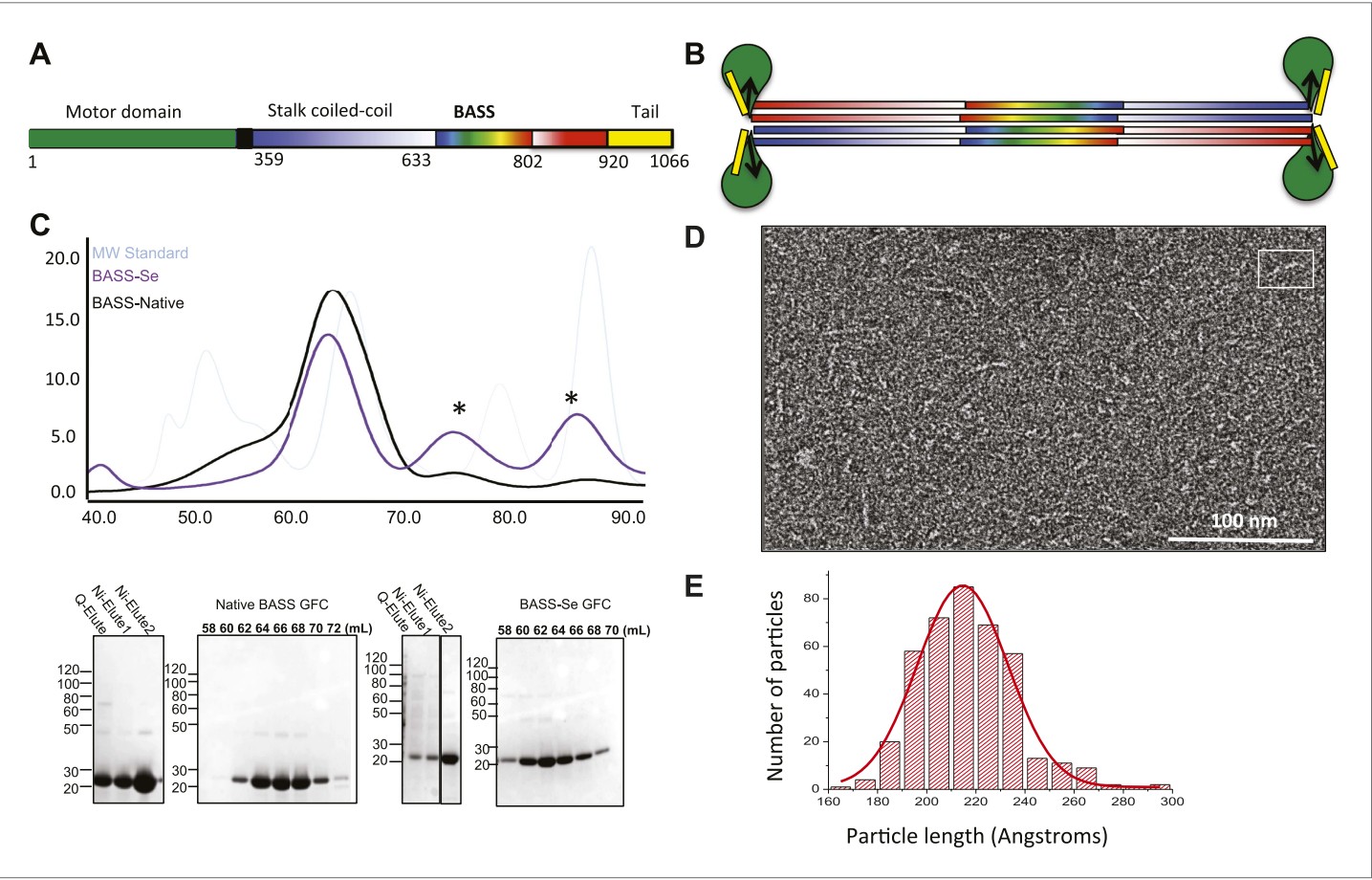

**Figure 1**. The Kinesin-5 BASS domain is an anti-parallel coiled-coil four-helix bundle that switches polypeptide partners at both ends. (**A**) Schematic domain structure of a Drosophila Kinesin-5 subunit (KLP61F). The bipolar assembly (BASS domain) is denoted by rainbow colors. The motor domain, N-terminal coiled-coil domain, the C-terminal helical domain, and tail domain are shown in green, blue, red, and yellow respectively. (**B**) Schematic of the Kinesin-5 tetramer. (**C**) Upper panel: Gel filtration (also known as, size exclusion or molecular sieve) chromatography (GFC) of BASS and Selenium-substituted BASS (BASS-Se); lower panel, purification steps of BASS and SDS-PAGE of GFC fractions. *Table 2* describes measured hydrodynamic properties of wt-BASS protein. (**D**) Negative stain electron microscopy (EM) of BASS tetramers. (**E**) Statistical analysis of BASS lengths measured using negative stain EM images describes an average length of 220 Å.

The following figure supplements are available for figure 1:

**Figure supplement 1**. Views of the BASS crystal unit cell displaying the packing of two BASS-dimer asymmetric units within the *P*6₄22 unit cell.

**Table 1.** Crystallographic statistics table for Kinesin-5 KLP61F BASS domain structure Determination.

| | BASS Native | BASS SeMet (Peak) | BASS-Hg (Peak) |
|---|---|---|---|
| Data collection | | | |
| Resolution range (Å) | 40.024–2.6 (2.74–2.6)* | 60.219–2.9 (3.06–2.90)* | 83.2–3.8 (3.87–3.80)* |
| Space group | P 6₄ 2 2 | P 6₄ 2 2 | P 6₄ 2 2 |
| Wavelength (Å) | 0.9795 | 0.9792 | 1.007 |
| Unit cell (Å): a, b, c | 138.65, 138.65, 105.79 | 139.07, 139.07, 104.35 | 139.57, 139.57, 100.94 |
| Total reflections | 155332 | 152078 | 65646 |
| Unique reflections | 18960 {15885}† | 13704 {12639}† | 5973 |
| Average mosaicity | 0.52 | 1.09 | 1.10 |
| Anomalous Multiplicity | – | 6.0 (5.6)* | 6.2 (5.3)* |
| Multiplicity | 8.2 (8.5)* | 11.1 (10.6)* | 11.0 (10.2)* |
| Anomalous Completeness (%) | – | 100.0 (100.0)* | 97.6 (99.3)* |
| Completeness (%) | 99.9 (100.0) {83.8}† | 100.0 (100.0) {92.4}† | 98.8 (99.4)* |
| $<I/\sigma (I)>$ | 10.8 (2.3)* | 11.9 (3.5)* | 18.3 (1.9)* |
| $R_{merge}$‡ | 0.099 (0.99)* | 0.11(0.68)* | 0.091 (0.73)* |
| Structure refinement | | | |
| $R_{work}$ | 0.22 (0.28)* | 0.24 (0.30)* | – |
| $R_{free}$ | 0.25 (0.37)* | 0.27 (0.33)* | – |
| Molecules per asymmetric unit | 2 | 2 | – |
| Number of atoms | 2355 | 2518 | – |
| Protein residues | 288 | 318 | – |
| Number of water molecules | 60 | – | – |
| RMS bond lengths (Å) | 0.006 | 0.006 | – |
| RMS bond angles (°) | 0.86 | 0.89 | – |
| Ramachandran favored (%) | 99.3 | 98.4 | – |
| Ramachandran outliers (%) | 0.0 | 0.0 | – |
| Clashscore | 5.5 | 5.6 | – |
| Mean B values (Å²) | | | |
| Overall | 80.6 | 86.7 | – |
| Main-chain atoms | 77.6 | 83.6 | – |
| Side-chain atoms | 84.3 | 89.8 | – |
| Solvent | 55.5 | - | – |

*Numbers represent the highest-resolution shell.

†Numbers represent the truncated data after treated with ellipsoidal truncation and anisotropic scaling.

‡$R_{merge} = \Sigma_{hkl}\Sigma_i|I_i(hkl) - I_{av}(hkl)|/\Sigma_{hkl}\Sigma_i I_i(hkl)$.

stabilized by sequential, alternating hydrophobic and ionic residue interfaces, each occupying three-helical turns (*Figure 2*). The resulting four-helix bundle, which is 22 Å wide, 26 Å high, and 260 Å long (*Figure 2A,B*) is the longest helical minifilment structure determined to date, and it displays a striking organization. It consists of a 220-Å-long central bundle that is visible by EM and matches fairly well to other anti-parallel four-helix bundles, where helical axes are separated by equal distances (*Figure 1D–E*, *Figure 3*, *Figure 4C*, panel III) but then transitions to an asymmetric diamond-shaped bundle near its ends, where C-terminal helical axes become displaced and N-terminal helical axes are closer mediated by helical bends, termed elbows (*Figure 4*, *Video 1*). The diamond shape helical organization at the poles brings the N-terminal ends of the helices at each pole into close juxtaposition so they emerge as parallel coiled-coil dimers, which were not resolved by EM (*Acar et al., 2013*), with a 90° rotational offset with respect to each other (*Figure 4*). This 26-nm BASS domain lies at the center

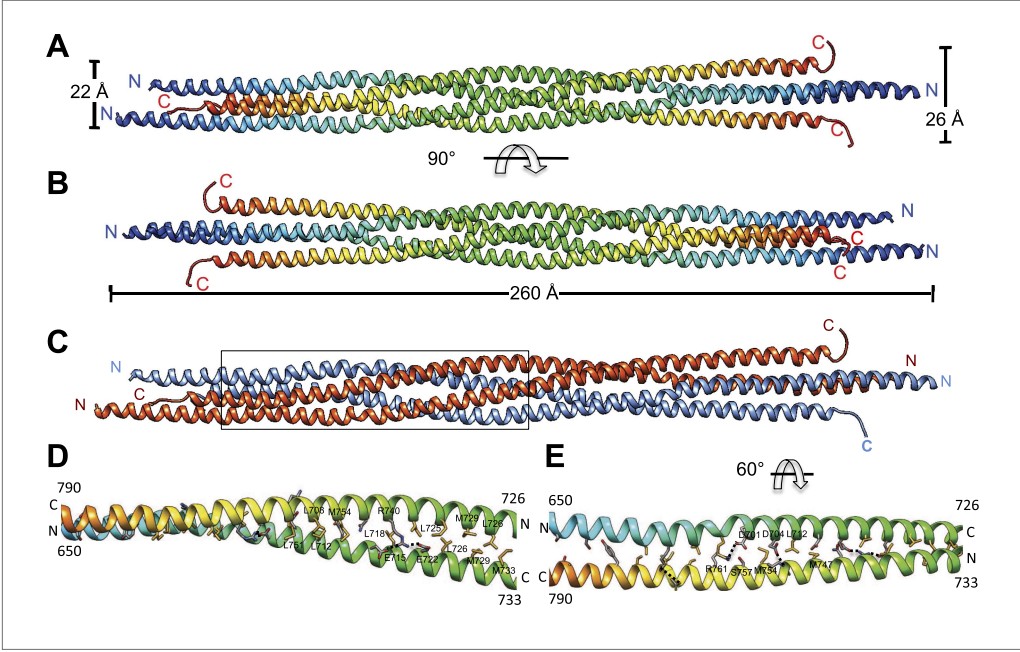

**Figure 2**. The crystal structure of the Kinesin-5 BASS domain tetramer: (**A**) Side view of the crystal structure of the KLP61F BASS tetramer (residues 640–796 shown) colored in rainbow, starting with N-termini in blue traversing to C-termini in red, respectively. Four monomers pack as anti-parallel pairs of anti-parallel coiled-coil dimers. (**B**) shows the BASS structure rotated 90° around the filament axis relative to panel **A**. The dimensions of the BASS tetramer bundle structure are shown. (**C**) Side view of the BASS tetramer, with two BASS anti-parallel dimers colored in blue and red, respectively. (**D**) Detailed interaction between two monomers in the BASS anti-parallel dimer. (**E**) A 60° rotated view of **D**.

of the rigid 60-nm Kinesin-5 central rod and can direct the assembly of four KLP61F subunits into a bipolar Kinesin-5 tetramer with pairs of N-terminal motor domains on each end capable of processive hand-over-hand motility (*Figure 4*; see discussion in section entitled 'Model for the Kinesin-5 minifilament' below). The unique anti-parallel BASS organization presented here is structurally novel as it does not match the initial predictions of an anti-parallel arrangement of parallel coiled-coils across the central helical rod region (*Kashina et al., 1996a*; *Acar et al., 2013*), yet it is entirely consistent with all previous data on the BASS domain (*Acar et al., 2013*). The BASS structure may be representative of a new class of force-pliable four-helical anti-parallel junctions capable of orienting two dimeric coiled-coils in opposite directions.

## Structure of the BASS central bundle

Alternating hydrophobic and ionic interfaces specify the assembly of the four BASS monomers to form a rigid central bundle. Dimeric anti-parallel coiled-coils are stabilized by complementary hydrophobic residues between *a* and *d* positions of the heptad repeat that are guided by salt bridges in the *e* and *g* positions (*Figure 2C–E*). The tetrameric central bundle displays a precise and unusual pattern of alternating hydrophobic and ionic interfaces between the four helices, not previously observed in other anti-parallel coiled-coil bundles (*Figures 4, 5, 6*; *Ernst and Brunger, 2003*; *Yadav et al., 2005*; *Fujiwara and Minor, 2008*; *Neculai et al.*. This pattern can be clearly observed by electrostatic surface representation of the BASS tetramer (*Figure 5—figure supplement 1*). Each interface occupies approximately three alpha-helical turns on each helix within the bundle, starting at the center with interface A of the BASS tetramer and symmetrically extending toward both ends with paired interfaces B–F (*Figures 5 and 6*; *Videos 2 and 3*). The hydrophobic central interface (A) is bracketed by an ionic interface (B), another hydrophobic interface (C), and finally another ionic interface (D) positioned at the 'elbows' marking the transition between anti-parallel four-helix bundle and parallel coiled-coils at the ends (*Figure 5*; *Video 2*). Interface A (*Figure 5*, panel IV) is composed of Met729, 730 and 733 from two helices and Leu725 and 726 from the anti-parallel helix. In other Kinesin-5 orthologs,

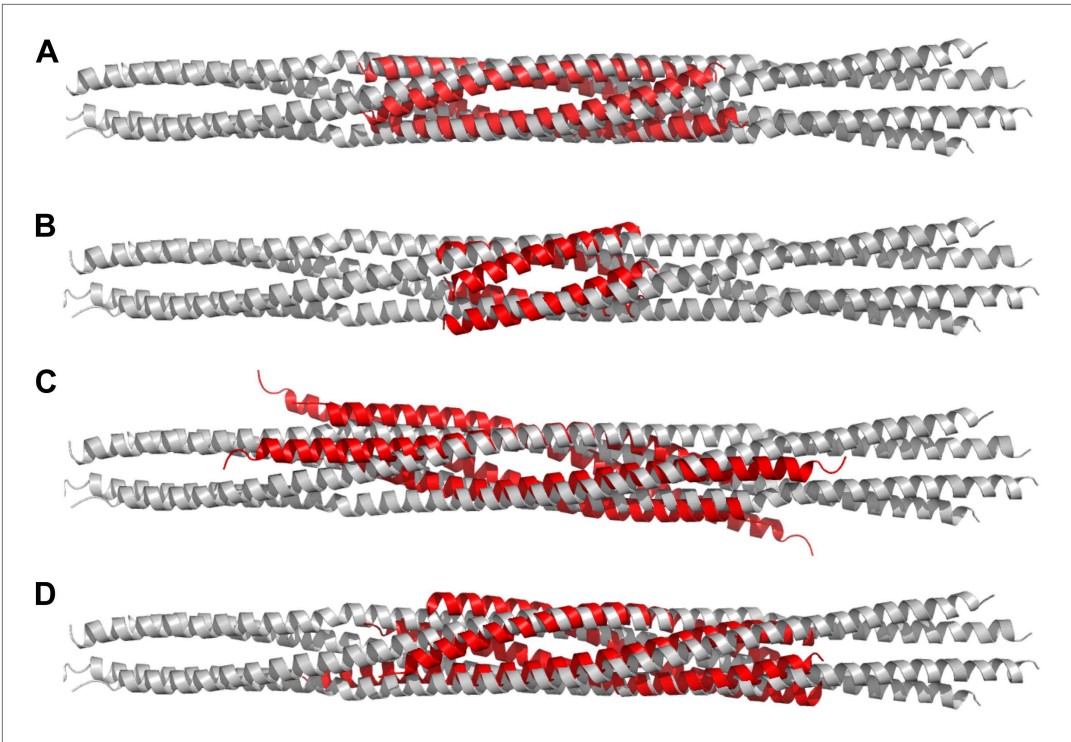

**Figure 3**. Structural comparisons of BASS structure with other four-helix bundles: BASS was compared to other four-helix bundles by superimposing α-carbon chains of these structures to the BASS structure using the lowest RMSD alignment in the program Pymol. Each of the following structures, shown in red, is superimposed onto the Kinesin-5 BASS, shown in gray: (**A**) Anti-parallel coiled-coil tetramerization domain of TrpM7 (PDB ID: 3E7K). (**B**) A GCN4-like designed anti-parallel coiled-coil (PDB ID: 1UNX). (**C**) A truncated neuronal SNARE complex (PDB ID:1N7S) (**D**) coiled-coil domain of tumor suspectibility gene 1 (TSG1) (PDB ID: 3IV1).

conservative substitutions of the cluster of methionine residues to other bulky hydrophobic residues are universally observed (*Figure 7*). Interface B lies outside interface A (*Figure 5*, panel III), and consists of salt bridges between Arg716 and Glu715 within a single chain, and Glu715/Glu722 and Arg740 residues of anti-parallel helices. Lys737 also engages the conserved Asp723 residues of the non-partner anti-parallel helices. Arg740 and Glu722 are highly conserved and are present in other species at equivalent positions or 3–4 residues apart positioning them on the same side of the helices in the bundle (*Figure 7*). These ionic interfaces are solution-accessible and contain visible electron density for several ordered water and cryoprotectant molecules. On the edges of interface B, hydrophobic heptad interfaces stabilize each BASS anti-parallel dimer, involving conserved Leu725 and 718 engaging the conserved Ile736. Next, the hydrophobic interface C is composed of Leu705, 708, and 712 in two helices engaging Met747, Leu751, and Met754 in anti-parallel helices (*Figure 5*, panel II). At the edge of interface C, complementary salt bridges are observed between Glu704 and 711 with His750 of the non-partner anti-parallel helices (*Figure 5*, panel II). Interface D lies at the elbow region, and involves two Arg761 residues engaging both Asp701 residues of the anti-parallel subunits at the center of the bundle (*Figure 5*, panel I). In total, these seven unique interfaces in the central bundle provide highly precise environments to form the tetramer (*Video 2*). Such a precise pattern of alternating complementary interfaces was not observed in other four-helical bundles, and is likely to be specifically required for the precise architecture of the Kinesin-5 bipolar minifilament. It may serve to mechanically stabilize the central rod's organization while transmitting forces between the two pairs of N-terminal motor domains at opposite ends of the tetramer. Moreover, the ionic interfaces B and D are open molecular pockets that are critical for specifying Kinesin-5 organization and are accessible to small molecules, such as the cryoprotectants observed in our structure. These pockets provide a new potential target site for small molecules that, instead of inhibiting the activity of the conserved kinesin superfamily motor domain in the manner of monastrol and related inhibitors (*Maliga et al., 2002*), may

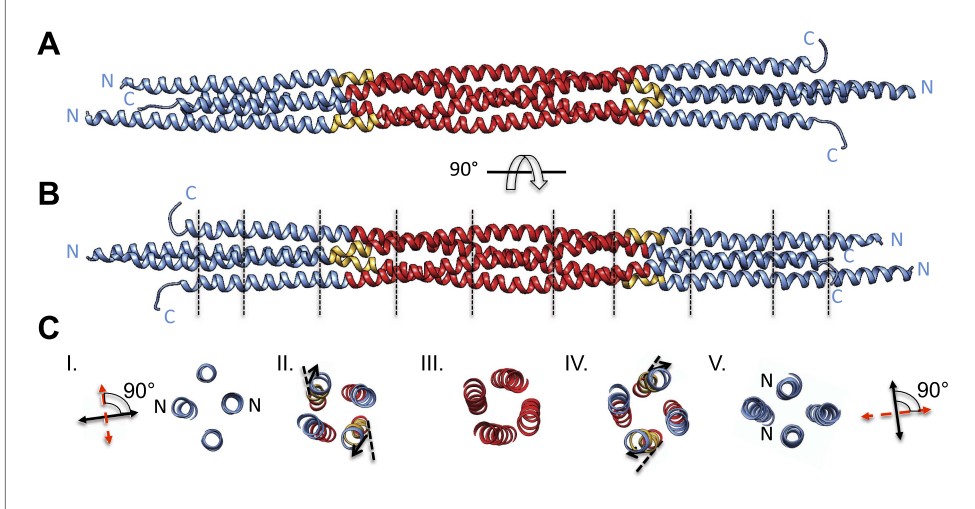

**Figure 4**. BASS tetramer consists of two regions with unique helical organizations. (**A** and **B**) Side view of the BASS structure, colored to mark two structural regions related by a dyad axis. The central bundle is shown in red. Two elbow regions (shown in gold) cause bends in N-terminal BASS helices to form the end regions. The end bundles, shown in blue, are asymmetric diamond-shaped four-helix bundles. The N-terminal helices are brought in close proximity to form parallel coiled-coils at the poles of the BASS tetramer, whereas the C-terminal helices are repositioned to be further away from the bundle center axis. Panel **B** is a 90-degree rotation of panel **A**. The lines in panel **B** represent regions where cross-section views of the structure are presented in part **C**. (**C**) Cross-section views of BASS using boundaries described in **B**. Panel I and V describe polar regions of the end bundles (note that N-terminal helices are closer together) and reveal their rotational offset by 90-degrees around the filament axis. Panels II and IV show the transition region between the central and end bundle regions with the elbows inducing a change in helical trajectory. Panel III shows a cross section of the central bundle region. Note that the helices in this region are fourfold symmetric around the central filament axis.

interfere with Kinesin-5 oligomerization by disrupting these salt bridges to inhibit Kinesin-5 biopolar tetrameric assembly and function during mitosis.

## The BASS domain N-terminal ends swap partners to form parallel coiled-coils at both ends

A second striking and unexpected feature revealed by the BASS tetramer structure is the transition from this anti-parallel four-helix central bundle to an asymmetric diamond-shaped four-helix bundle near its ends (*Figures 2, 4, 6* and *Video 3*). This transition promotes a swap in helical organizations, from anti-parallel coiled-coil dimers in the central bundle to parallel coiled-coil dimers at its ends. The swap transition is promoted by two helical 'elbows' in which Pro699 and Gly693 bend the helices by 10° which ultimately bring the N-terminal helices at each pole into close proximity, emerging as parallel coiled-coils associating directly through heptad residue packing on opposite ends with a 90° rotational offset around the filament axis (*Figures 2, 4 and 6*). Near the ends of the bundle, two interfaces mediate these helical swaps: a hydrophobic interface (E) and a di-tyrosine interface (F) (*Figure 6* and *video 3*). Interface E (*Figure 6*, panel III) maintains a 45° rotated anti-parallel four-helix bundle organization via Met687, Leu691, Leu694 of two helices packing with Met768 and Ile772 of the anti-parallel helices in a four-helix hydrophobic packing interaction (*Figure 4C*, panel II and V). Within interface F, C-terminal helices are positioned further away from center of the bundle axis, due to two Tyr residues whose rings undergo end-to-end hydrogen bonding between their hydroxyl groups. Two His683 residues of the N-terminal helices stack their imidazole rings against Tyr775 residues through π–π interactions to stabilize their end-on orientations (*Figure 6*, panel II). Bulky residues Tyr775 or His683 are conserved suggesting that this motif is retained with some modifications in all Kinesin-5 orthologs (*Figure 7*). Beyond interface F, the N-terminal helices swap organization to form parallel coiled-coils between residues 670–640 stabilized by hydrophobic *a* and *d* heptad packing interaction (pCC interface). Two heptad interactions are observed between the two N-terminal helices, with two half heptads at each N- and C-terminal end. Ile651, His658 and Met665 form the 'd' positions, and Asn655, Leu662, and Phe669 form the 'a' positions in these heptads. Phe669-Phe669 residues pack using π–π hydrophobic

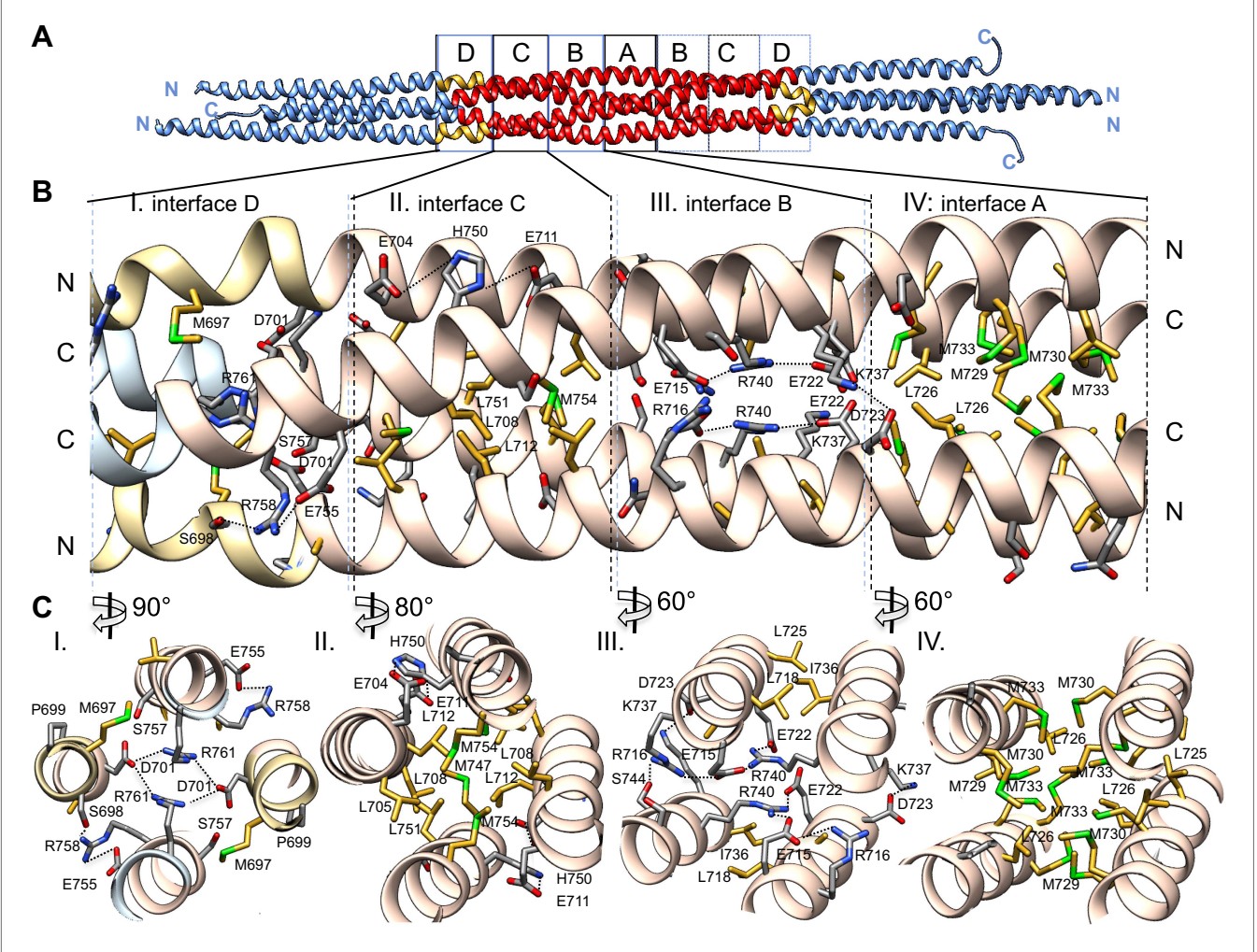

**Figure 5**. The BASS central bundle is assembled through an alternating pattern of antiparallel hydrophobic and ionic interfaces. (**A**) Full side-view of the BASS structure as shown in **Figure 4**, describing the regions of the BASS bundle, termed interfaces **A**–**D**. Interfaces B, C and D are twofold symmetric and extend outward from a single interface A. (**B**) Detailed side view of the left side of the central bundle region depicting interfaces D, C, B, and A, respectively. Panel (I), a side view of interface D: Residues Arg761 bind Asp701 from anti-parallel helices and Arg758 binds Glu755 and Ser698 of the non-partner helices. Panel (II), a side view of interface C: Leu705, Leu708, and Leu712 from two helices pack against Met747, Leu751, and Met754 of the anti-parallel helices, in a four-helical interface. Glu704, Glu711 form salt bridges with His750 of the anti-parallel helices away from the bundle axis. Panel (III), side view of interface B where Arg740 forms salt bridges with Glu715 and Glu722 of the anti-parallel helices. Lys716 forms salt bridges with Glu715, while Lys737 forms a salt bridge to Glu723 of the non-partner anti-parallel helices. Panel (IV), side view of interface A: Leu725 and Leu726 pack against Met729, Met730, and Met733 in four-way helical packing. (**C**) Detailed cross-section view of interfaces A, B, C, and D showing the same residues described above. Panels I–IV are cross sections of corresponding views shown in **B**, but rotated by either 60, 80 or 90° across the filament axis.

The following figure supplements are available for figure 5:

**Figure supplement 1**. Surface electrostatic potential view of BASS tetramer interfaces.

interactions that stabilize the C-termini to form parallel coiled-coil structures (**Figure 6B–C**, panel I). In Kinesin-5 sequence alignments, the last two heptads contain a Phe residue in either *a* or *d* position, which is likely to stabilize the reorganization at the tips of the N-terminal parallel coiled-coil structures via benzyl–benzyl ring interactions (**Figure 7**). This suggests that the N-termini of full-length Kinesin-5 extending beyond the boundaries of the BASS tetramer likely form continuous parallel coiled-coils.

## Testing the Kinesin-5 tetrameric model using structure-based mutagenesis
To test the role of the conserved residues within interfaces A–F and the swapped parallel coiled-coils in the assembly of the BASS domain, we disrupted them by structure-based mutagenesis and assessed

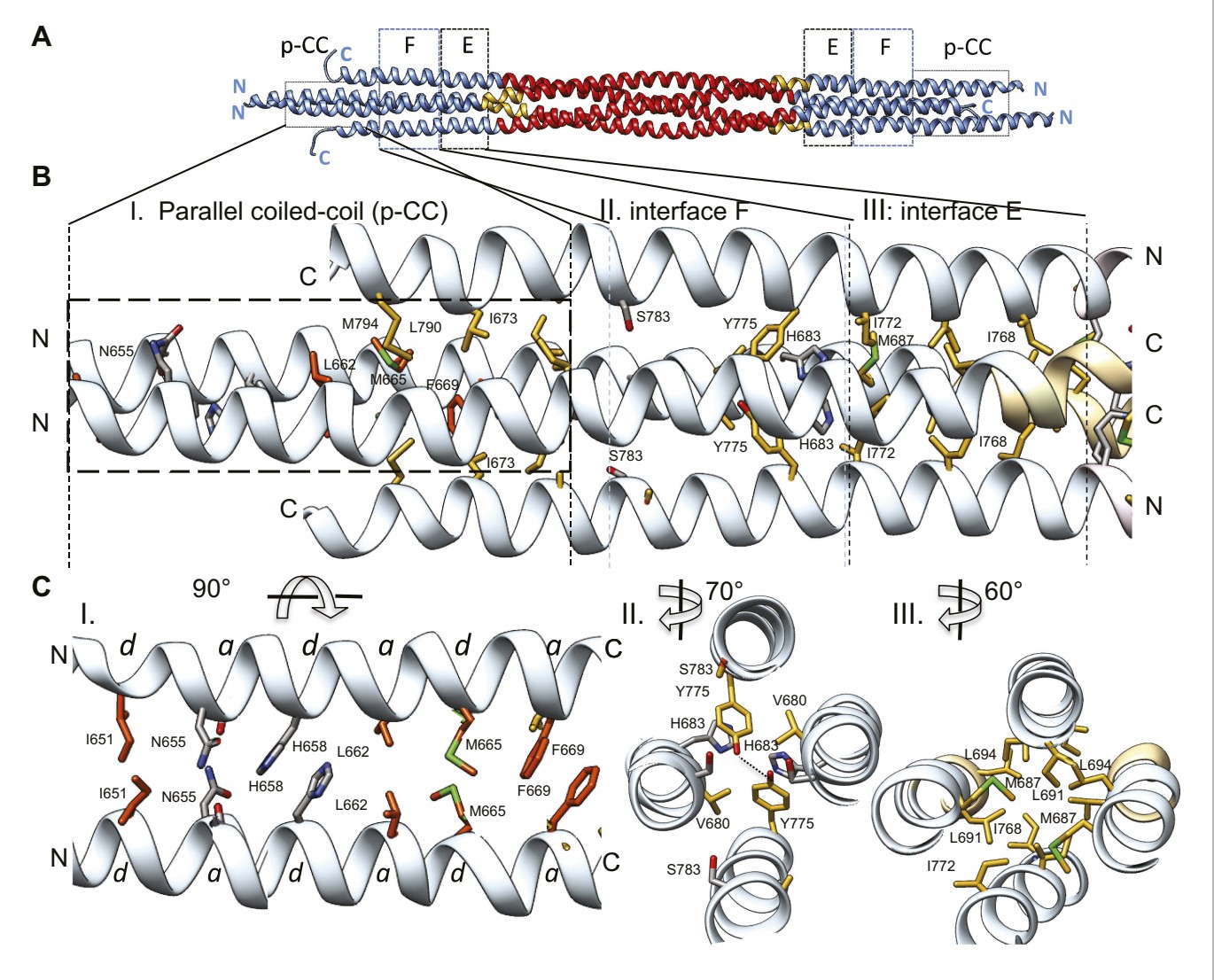

**Figure 6**. The tetrameric BASS domain N-terminal ends swap partners to form parallel coiled-coils at the bipolar filament ends. (**A**) Full side-view of the BASS structure as shown in ***Figure 4***, describing the regions of the BASS bundle. Interfaces E, F and the parallel coiled-coil interface (p-CC) are marked. (**B**) Detailed view of the end bundle interfaces showing the left side bundle of the BASS tetramer including the p-CC and interfaces F and E, respectively. Panel (I), side view of the parallel coiled-coil interfaces formed by residues in the N-terminus of BASS (residues 650–6700). Residues mediating a heptad repeat hydrophobic interaction are shown in orange and include Ile651, Leu662, Met665, and Phe669. C-terminal helices also bind this region using hydrophobic interfaces. Panel (II), side view of the interface F: Tyr775 binds Tyr775 through an end-to-end ring packing, supported by His683 π–π packing against the Tyr775 ring residue. This interface positions the C-terminal helices further away from the bundle axis. The remainder of the helical bundle contains small or non-interacting residues such as Ser783. Panel (III), side view of interface E: residues Met687, Leu691, and Leu694 of two helices packed against Ile768 and Ile772 of the anti-parallel helices in a four-helical bundle interface. (**C**) Top-to-bottom and cross-section views of the end bundle interfaces. These views are rotated by the angle described from views shown in part **B**, panels I, II, III, respectively. Panel I shows a top-to-bottom view of the parallel coiled-coil of two N-terminal helices. The heptad interactions are marked a and d. In total, the 'a' and 'd' positions of three heptads are observed. Phe669 packs against Phe669 to stabilize the helical 'swap' in this region. Panel II is a cross-sectional view of interface F rotated 70°. Panel III shows a cross-sectional view of interface E rotated 60°.

the oligomeric state of the resulting BASS mutant proteins using hydrodynamic analyses (***Figure 8***; ***Tables 2 and 3***) (***Acar et al., 2013***). As expected (***Acar et al., 2013***), wild-type BASS forms a mono-disperse tetramer based on gel filtration chromatography and sucrose density gradient centrifugation (***Figures 1 and 8B***). Mutations that target residues within various hydrophobic and ionic interfaces in the BASS central bundle cause defects in BASS oligomerization, observed as a shift in the protein's elution profile from tetramers (broken vertical line) to monomers (solid line) or an intermediate,

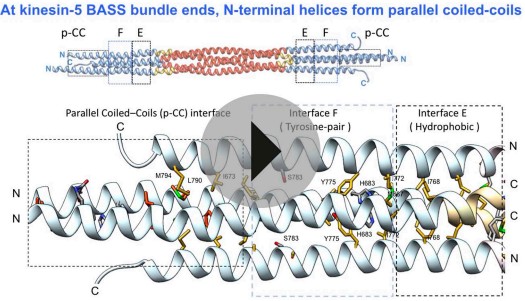

The helical organization of the Kinesin-5 BASS Bipolar tetrameric Bundle

**Video 1**. Structural organization and novel fold of BASS tetramers. This video accompanies *Figure 4*.

The Kinesin-5 BASS central four-helix bundle is organized by alternating and symmetric hydrophobic and ionic interfaces

Red : negative charge
Blue: positive charge
Yellow: hydrophobic

**Video 2**. The Kinesin-5 four-helical bundle is organized by alternating and symmetric hydrophobic and ionic interfaces. This video accompanies *Figure 5*.

At kinesin-5 BASS bundle ends, N-terminal helices form parallel coiled-coils

Parallel Coiled–Coils (p-CC) interface     Interface F ( Tyrosine-pair )     Interface E ( Hydrophobic )

**Video 3**. At the Kinesin-5 BASS bundle ends, the N-terminal helices form parallel coiled-coils. This video accompanies *Figure 6*.

presumptively dimeric species (*Tables 2 and 3*; *Figure 8*). Strikingly, single Arg740 and Arg761 residue mutations in interfaces B and D of BASS cause potent defects in specifying BASS tetramers producing a mixture of dimer and tetramer species (*Figure 8H–I*). In contrast, mutagenesis of the region where helix-swapping forms parallel coiled-coils at the ends of the BASS domain (Phe669Glu BASS mutant) remained tetrameric (*Figure 8D*; *Tables 2 and 3*). Mutagenesis of multiple interfaces across the tetramer, for example interfaces A and F (*Figure 8J–K*); interfaces, A, F and the parallel coiled-coil swap regions (*Figure 8L*); and interfaces A, B, C, D, F and the swap region (*Figure 8M*) enhanced the disruptive effects of the single mutants and produced monomeric BASS species (*Tables 2 and 3*). Our mutational analysis validates the proposed BASS tetrameric organization revealed by the structure by demonstrating the critical roles of the alternating interfaces in forming and stabilizing the BASS tetramer and further suggests that both hydrophobic and ionic interfaces have cumulative guiding roles in selectively organizing the BASS tetramer fold and stabilizing the Kinesin-5 tetramer minifilaments. The strong defects observed for single Arg BASS mutants (Arg740 and 761) suggest that small molecules that interfere with these ionic interactions in BASS interfaces B and D can disrupt Kinesin-5 minifilament assembly and may inactivate Kinesin-5 sliding motility.

## Model for the Kinesin-5 minifilament

To investigate how the BASS domain influences the assembly and structure of Kinesin-5 into minifilaments, we built a model of the Kinesin-5 central rod minifilament based on the structure of the BASS tetramer, Kinesin-5 sequence parameters, and the overall length of the 79-nm Kinesin-5 tetramer based on EM (*Figure 9*; Materials and methods') (*Kashina et al., 1996a*; *Acar et al., 2013*). Our model was built by superimposing dimeric parallel coiled-coils at the bipolar ends of the BASS tetramer structure (*Figure 9*) and indicates that; (i) the rigid tetrameric fold of the Kinesin-5 BASS domain into bipolar tetramers dictates long-range strict structural and functional assembly of helical coiled-coils N-terminal to BASS with the helical section C-terminal to BASS (*Figure 9*). We modeled the Kinesin-5 motor and tail domain to lie at the ends of the central rod, consistent with EM and with them both binding MTs (*Acar et al., 2013*; *van den Wildenberg et al., 2008*). The anti-parallel organization of BASS dimers likely produces head-to-tail junctions at the bipolar ends of Kinesin-5 coupling two dimeric kinesin heads with the conserved C-terminal tails, by traversing in the opposite orientation within the central rod filament; and (ii) The Kinesin-5 sequences C-terminal to BASS may not be fully helical nor form coiled-coil structures because their length is shorter than the length of the N-terminal-to-BASS domain coiled-coils (*Figures 1, 9*). Moreover, while the overall model length matches that seen by EM, the rod itself is longer than that observed, similar to the Kinesin-1 rod (*Acar et al., 2013*). Further high-resolution structural studies of the longer regions of the Kinesin-5 rod filament lying outside the BASS and motor domains (*Bodey et al., 2009*) will be required to further refine the full Kinesin-5 bipolar tetramer model.

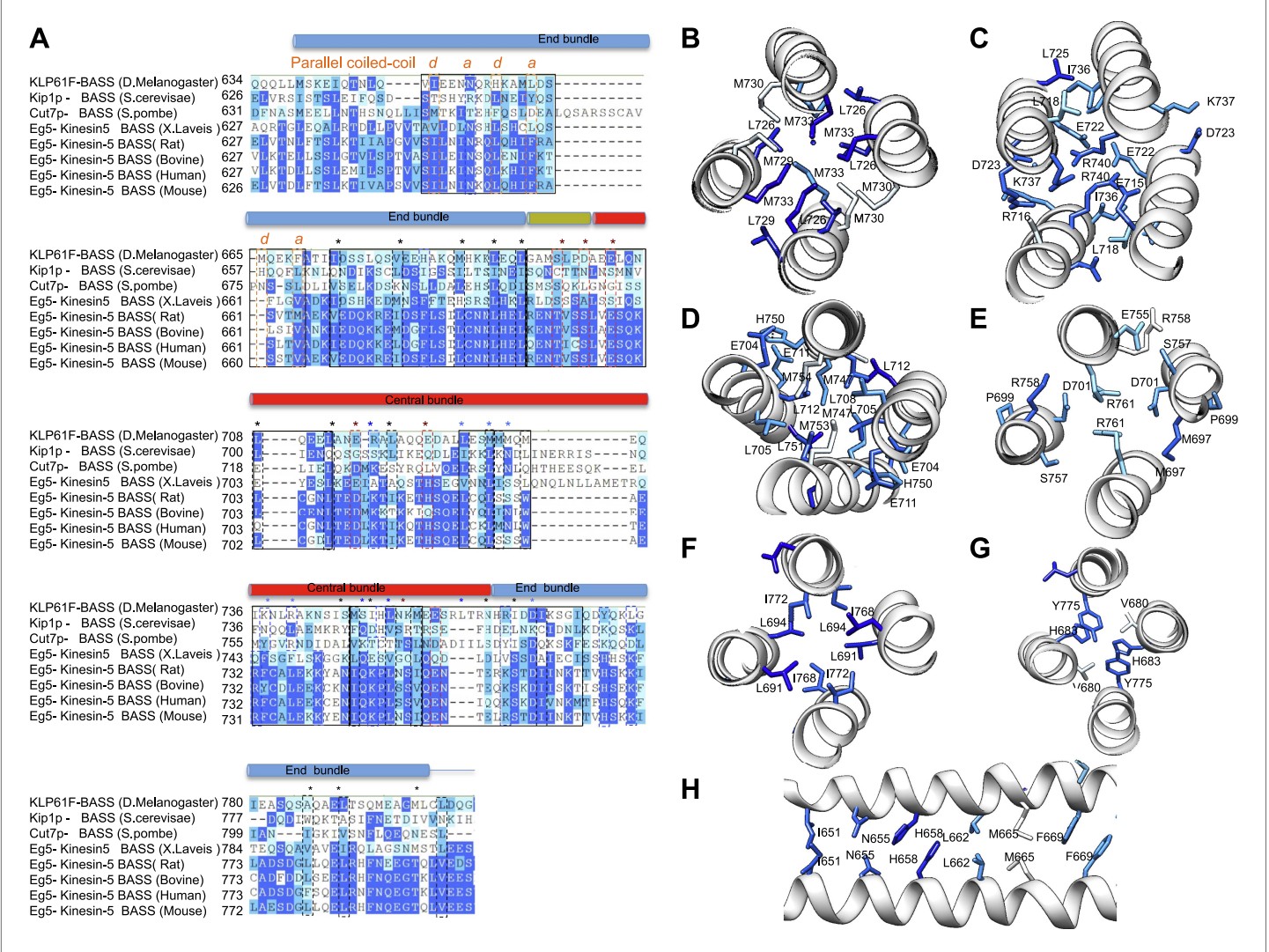

**Figure 7**. BASS structure features are conserved across Kinesin-5 family: (**A**) Sequence conservation between the KLP61F BASS sequence and other Kinesin-5 orthologs. The alignment shows that many hydrophobic and ionic interfaces (depicted in *Figures 5 and 6* as **A–F**) are conserved and include minor positional variations that are preserved at similar positions of the helices. (**B–H**) Structural views of interfaces **A–F** (described in *Figures 5 and 6*) with sequence conservation mapped on the structure in colors corresponding to those displayed in panel **A**.

## Concluding remarks

The BASS domain tetramer structure is, to our knowledge, the first atomic structure of a kinesin super-family oligomerization domain determined to date. The structure explains how Kinesin-5 assembles into bipolar, homotetrameric minifilaments, but what are the biological implications of the BASS domain structure? Our main findings are that; (i) the central bundle of the BASS tetramer is a novel anti-parallel coiled-coil four-helix bundle, which is held together by a unique and alternating arrangement of hydrophobic and ionic interfaces; and (ii) helical bends, at the outer ends of the central bundle promote transitions into parallel coiled-coils, which are offset by 90° around the filament axis with respect to each other. One significant consequence of this structure is that the BASS tetramer orchestrates a parallel organization and rotational offset of the pairs of motor domains at each end of a Kinesin-5 minifilament resembling the organization of the two motor domains comprising the motile end of the unipolar Kinesin-1 motor. This organization may be important in allowing the adjacent motor domains on either end of a Kinesin-5 minifilament to step processively in a hand-over-hand fashion along each MT (*Vale and Milligan, 2000*; *Valentine et al., 2006b*; *Kaseda et al., 2009*), thus enabling the Kinesin-5 tetramer to slide MTs outward or to constrain their sliding apart within the

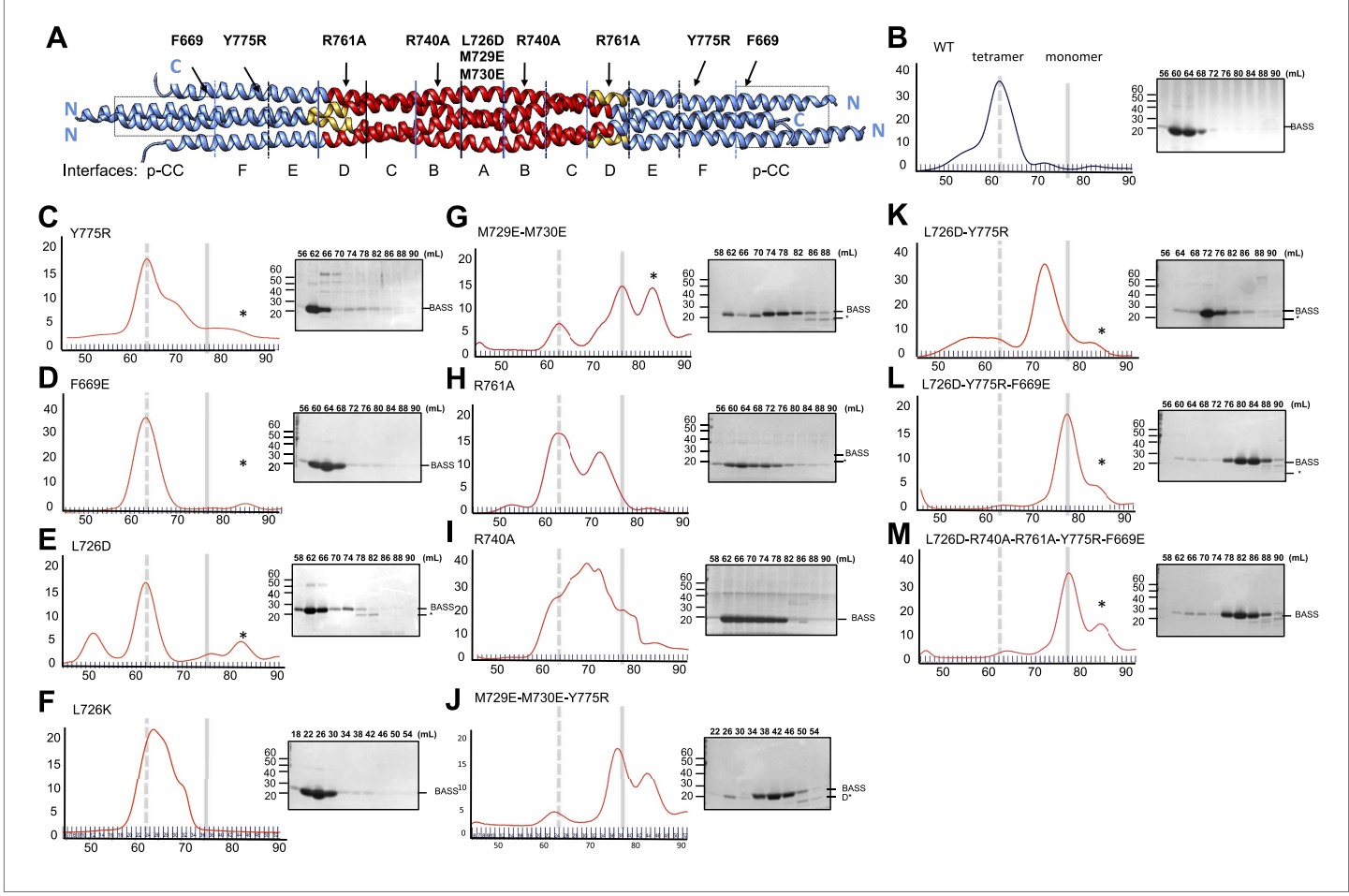

**Figure 8**. Structure-based biochemical analysis of the BASS interfaces in stabilizing Kinesin-5 bipolar minifilaments. (**A**) Schematic view of BASS tetramer, shown in *Figure 4*. The model is divided into zones marking each of the interfaces described in *Figures 5 and 6*. Mutated residues are described above the model, and the interfaces described in each region are described below the model. Each of the panels below (**B**–**M**) includes a gel filtration chromatography elution profile on the left in which the tetramer and monomer peaks are indicated by broken and solid lines, respectively. An SDS-PAGE of the column fractions marked by volume (mLs) is shown on the right. A BASS degradation peak is observed under some conditions: (**B**) Wt: remains mostly tetrameric (broken line). (**C**) Tyr775Arg: mainly a tetrameric (broken line), with moderate amount of intermediate peak ahead of monomer peak (solid line). (**D**) Phe669Glu is a tetramer. (**E**) Leu726Asp: mainly a tetrameric peak (broken line), with a small intermediate peak, ahead of monomer peak (solid line). (**F**) Leu726Lys: mainly a tetrameric peak (broken line), with a small intermediate peak, ahead of monomer peak (solid line). (**G**) Met729Glu/Met730Glu: very low tetramer peak (broken lines) and mostly monomer peak. (**H**) Arg761Ala: mixture of tetramer peak (broken line) and intermediate peak ahead of monomer peak (solid line). (**I**) Arg740Ala: mixture of tetramer peak (broken line) and intermediate peak ahead of monomer peak (Solid line). (**J**) Met729Glu/Met730Glu/Tyr775Arg: very low tetramer peak (broken lines) and mostly monomer peak. (**K**) Leu726Asp/Tyr775Arg: little tetramer peak (broken lines) and mainly intermediate peak between tetramer and monomer peak (solid line). (**L**) Leu726Asp/Tyr775Arg/Phe669Glu: almost no tetramer peak (broken lines), co-eluting with monomer peak (solid line). (**M**) Leu726Asp-Arg740Ala-Arg761Ala-Tyr775Arg-Phe669Glu: no tetramer peak (broken lines) and almost completely monomer (solid line). *Table 3* summaries the hydrodynamic properties of BASS mutants described here. *Table 2* describes hydrodynamic analysis for mutants shown in panels **D** and **M**.

mitotic spindle (*Kapitein et al., 2005*; *van den Wildenberg et al., 2008*; *Hentrich and Surrey, 2010*; *Weinger et al. 2011*). A second consequence is that the four helices of the BASS domain are highly intertwined, in a manner reminiscent of the plectonemic coiling of the two polynucleotide chains of a DNA double helix, far more so than the simple association of two anti-parallel side-by-side dimeric coiled-coils proposed previously (*Kashina et al., 1996a*; *Acar et al., 2013*). The resulting heavily intertwined four-helix bundle is likely to be torsionally stiff, and this may be crucial in setting the 90° offset between the two dimeric ends of the BASS structure. This offset could in turn contribute to the motor's observed threefold preference for crosslinking anti-parallel vs parallel MTs (*van den Wildenberg et al., 2008*) but further work is required to test this idea. We hypothesize that the nature of the bundling of the BASS domain is critical for the Kinesin-5-mediated sliding filament mechanism because it functionally

**Table 2.** Hydrodynamic properties of BASS and its mutants*

| Protein | R-stokes | S-Value | MASS/Oligomer |
|---|---|---|---|
| wt | 5.52 | 4.00 | 92 kDa (tetramer) |
| F669E | 5.62 | 4.25 | 102 kDa (tetramer) |
| L725D-R740 A-R761A-Y775R-F669E | 3.77 | 2.07 | 32 kDa (monomer) |

*Masses were calculated from a combination of gel filtration chromatography and sucrose density gradient sedimentation. Using these approaches, the Stokes Radii (R-Stokes) and sedimentation value (S-values) were determined for each protein and used to calculate the mass.

**Table 3.** Hydrodynamic properties of BASS and its mutants shown in *Figure 8*

| BASS protein | Interface | Elution peaks | Oligomerization state |
|---|---|---|---|
| wt | None | 62.5 ml | Tetramer* |
| L726D | A | 62.5 ml | Tetramer |
| L726K | A | 62.5 ml | Tetramer |
| F669E | p-CC | 62.5 ml | Tetramer* |
| Y775R | F | 62.5, 71 ml | Tetramer/dimer |
| R761A | D | 62.5, 71 ml | Tetramer/dimer |
| R740A | B | 62.5, 71, 77 ml | Tetra/dimer/monomer |
| M729E-M730E | A | 62.5, 77 ml | Monomer |
| L726D-Y775R | A, F | 71 ml | Dimer |
| M729E-M730E-Y775R | A, F | 80 ml | Monomer |
| L726D-Y775R-F669E | A, F, p-CC | 80 ml | Monomer |
| L726D-R740 A-R761A-Y775R-F669E | A, B, D, F, pCC | 80 ml | Monomer* |

*The masses of these proteins were measured as described in *Table 2*.

links the pairs of motor domains located at opposite ends of the Kinesin-5 minifilament by transmitting forces and possibly allowing allosteric signaling between them (*Figure 10*). Thus, we propose that the alternating pattern of interfaces revealed by the BASS structure serves not only to specify the assembly of Kinesin-5 subunits into bipolar tetramers, but also to increase central rod stability during force transmission in a manner that facilitates its ability to push apart or to restrain the sliding apart of cross-linked MTs. Measuring the magnitude of forces that the rod can transmit or withstand would be a useful test of this idea, but this is currently not technically feasible. Finally, we note that we have obtained no convincing evidence for the assembly of Kinesin-5 tetramers into higher order oligomers, and the extensive inter-twining of its four chain makes it's dissociation into, for example, dimers during force transmission unlikely. Thus it seems reasonable to propose that the Kinesin-5 bipolar tetramer represents the functional unit of Kinesin-5 activity, consistent with motility assays (*Kapitein et al., 2005*; *van den Wildenberg et al., 2008*; *Weinger et al., 2011*).

The conservation in amino acid sequence, structural features and length of the 4-stranded helical rod among members of the Kinesin-5 family, but not in members of other kinesin families, suggests that the properties of the BASS domain reported here are likely to be key to understanding how Kinesin-5 can either drive or constrain spindle pole separation during spindle assembly, spindle maintenance, and anaphase B spindle elongation in a broad range of organisms (*Enos and Morris, 1990*; *Kapitein et al., 2005*; *Saunders et al., 2007*; *van den Wildenberg et al., 2008*; *Brust-Mascher et al., 2009*). The bipolar organization of Kinesin-5 resembles that of the bipolar class 2 myosin filament that drives the sliding filament mechanism of muscle contraction, cytokinesis and other forms of motility in non-muscle cells (*Huxley, 1963*; *Turbedsky et al., 2005*; *Billington et al., 2013*). However, whereas varying numbers of myosin-2 motors can assemble into filaments of variable length, all capable of

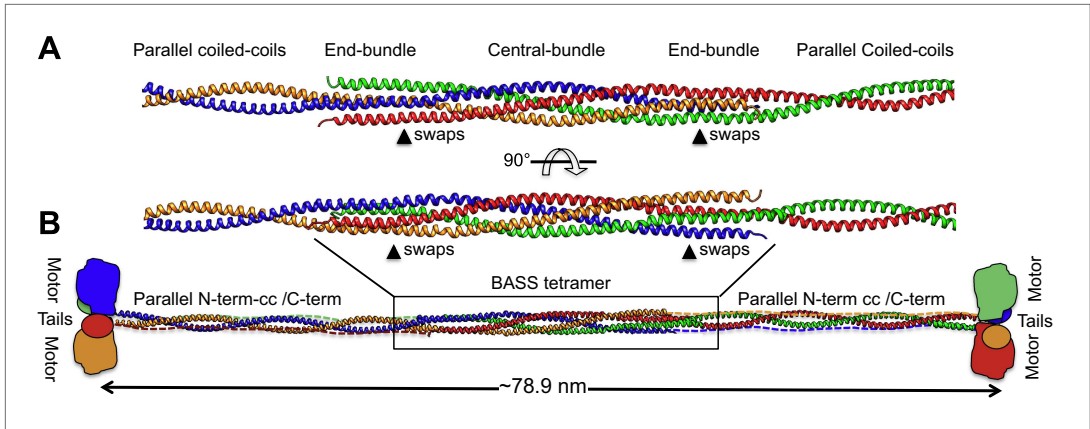

**Figure 9**. Modeling the Kinesin-5 tetramer minifilament. (**A**) Model of Kinesin-5 central rod coiled-coil junction. Parallel coiled-coil structures were fit to the poles of the BASS tetramer by superimposing alpha carbons. The regions where the structures swap organization from an anti-parallel coiled-coil bundle to a parallel coiled-coil dimer are marked by arrowheads (swaps). (**B**) Cartoon of a full-length Kinesin-5 minifilament based on a model for the rod structure showing the central role of the BASS tetramer in organizing the N-terminal coiled-coil registers and positioning the C-terminal region to fold onto the N-terminal coiled-coil filament. The Kinesin-5 N-terminal motor and C-terminal tail domains, both bind MTs, are organized through long range folding of the BASS tetramer at the center of the Kinesin-5 rod.

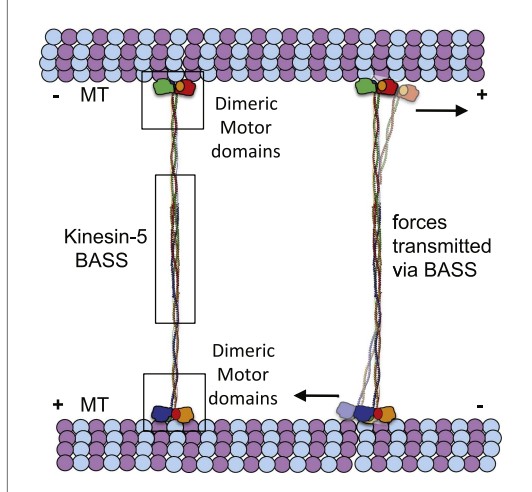

**Figure 10**. The implications of BASS structure on the Kinesin-5 motility and force transfer mechanism. Schematic model of Kinesin-5 minifilament showing the potential role of BASS in force transfer between two motile ends of Kinesin-5 tetramers: the orientation of Kinesin-5 tetramers and role of BASS bipolar tetramer in transmitting the forces between two motile Kinesin-5 ends.

pulling actin filaments inward, we propose that the unique structure of the BASS domain allows it to organize four Kinesin-5 subunits into stable, mechanically robust bipolar tetramers of uniform length, that represent functional units capable of bearing both compressive and tensile forces as they slide cross-linked MTs outward, or resist their outward sliding. Finally, we speculate that our structure may enable the development of small molecule inhibitors targeting the ionic interfaces within the BASS central bundle, leading to the disruption of Kinesin-5 tetramer assembly and inhibiting its function in rapidly proliferating cancer cells, thereby providing a possible avenue towards therapeutic intervention (*Owens, 2013*).

## Materials and methods

### Kinesin-5 BASS construct design, expression and purification

Oligonucleotide primers for shortened KLP61F (*Drosophila* Kinesin-5) wt construct (residues 633–835) were designed based on hydrodynamic and EPR analysis (*Acar et al., 2013*). PCR and isothermal assembly was used to build a bacterial expression BASS construct with a C-terminal His tag and was confirmed by sequencing. Expression was performed in SoluBL21 *Escherichia coli* strain in 6–12 liter formats and expression induced with 0.5 mM IPTG overnight at 18°. Selenomethionine substituted (Se-BASS) BASS protein was expressed in soluBL21 *E. coli* strain using a metabolic labeling strategy, where growth and expression were performed using minimal media containing all amino acids but with selenomethionine replacing Met (*Van Duyne et al., 1993*). BASS mutants were expressed and purified using the same methods as native BASS. Overlapping poly-oligonucleotide DNA synthesis strategy (Life technologies, GeneArt, Germany)

was used to generate structure-based BASS mutant constructs, which were assembled with iso-thermal assembly and expressed using the strategy described above.

BASS-containing bacterial pellets were lysed using a micro-fluidizer in (300 mM KCl, 50 mM HEPES, 1 mM MgCl$_2$, 3 mM β-mercaptoethanol with protease inhibitors including 1 mM PMSF, 1 μg/ml leupeptin, 20 μg/ml benzamidine, and 40 μg/ml Nα-p-Tosyl-L-Arg). The bacterial lysate was clarified by centrifugation at 18k rpm for 30 min at 4°C. Ni-NTA affinity was used to purify BASS, and passage over HiTrap Q HP cation exchange in low salt (70 mM KCl, 50 mM HEPES, 1 mM MgCl$_2$) was used to remove contaminants where BASS eluted in the flowthrough. A second Ni-NTA affinity step was used in conjunction with 30K Amicon Filters to concentrate the BASS. The concentrated BASS tetramer was applied on a HiLoad 16/600 Superdex 200 gel filtration column using an AKTA Purifier (GE Healthcare) and fractions were analyzed using SDS-PAGE (*Figure 1B*). The purified BASS protein was concentrated and used immediately for crystallization, EM and hydrodynamic analysis, or was frozen in liquid nitrogen.

### Hydrodynamic mass measurement analysis

The mass and oligomerization state of wt or mutant BASS proteins was determined as previously described (*Acar et al., 2013*). Briefly, 0.2 mg protein mass standards (Ovalbumin, BSA, and Aldolase) and 0.2 mg BASS protein were added at the top of 5–20% sucrose gradient prepared in Beckman 9/16*3.5 UC Tubes, and centrifuged in a Beckman Ultracentrifuge using a SW41–Ti rotor for 20 hr at 40K rpm. S-values for wt BASS or mutants were determined using a linear extrapolation from known standard S-values, calculated by identifying positions of standard and BASS proteins fractions using SDS-PAGE. S-values of BASS were combined with Stokes Radii measured using calibrated gel filtration columns (*Figure 1D*).

### Crystallization and X-ray diffraction

BASS tetramer was concentrated to 25 mg/ml for crystallization trials, which were performed using a mosquito robot (TTPlabtech) by mixing 100 nL drops with equal amounts of 2500 distinct solutions of home-made and commercial crystallization screens (Qiagen). Multiple crystallization conditions were identified; however, 0.1–0.2 μm hexagonal-shaped crystals were obtained and refined using 4–6% (+/−) 2-Methyl-2, 4-pentanediol (MPD), 100 mM MES pH 6–6.5. Crystals appeared in 1 week and grew to maximal size in 3 weeks. SeMet BASS protein also formed hexagonal crystals using similar conditions to native BASS. For cryo-protection, native- and SeMet BASS crystals were transferred to solutions with higher MPD concentrations or rapidly immersed in Paratone oil before freezing in liquid nitrogen by looping in nylon loops. 1 mM Mercury (II) nitrate was added to drop solutions containing native crystals to produce mercury-substituted crystals, which were cryo-frozen as described above. X-ray diffraction was screened using 96-position cassettes using robotic auto-mounting system at the Stanford Synchrotron Radiation Laboratory (SSRL) using either 14-1 or 12-2 beamlines. More than 150 native and 80 SeMet BASS crystals were screened.

### Structure determination, refinement, and model building

BASS native X-ray diffraction data were anisotropic at best, diffracting to 3.1 Å in one direction and 2.6 Å in the other directions. X-ray diffraction data were indexed with program imosfilm and scaled with Scala (*Project, 1994*), combined with ellipsoidal truncation and anisotropic scaling, which included 84% completeness in the overall resolution shells. We used 2.6 Å as the high-resolution cut-off to avoid excessive loss of completeness. Selenium edge X-ray diffraction data set was collected at 0.9792 Å wavelength. The diffraction was also anisotropic (minimum Bragg spacing 2.9 Å, and 3.1 Å in the weakest, intermediate, and strongest diffracting directions). We used 2.9 Å as the high-resolution cut-off with 92% completeness in the overall resolution shells. The data set was not adequate to provide phase information directly, due to high number of Met at the hexagonal sixfold axis. Phase information was initially determined using the single anomalous dispersion (SAD) method, using a 3.8 Å Mercury diffraction data set collected at the Mercury anomalous edge at 1.007 Å wavelength, with the program SHELEXD (*Sheldrick 2010*). Five heavy atom sites were identified and refined with SHELEXD. Initial BASS electron density was used to build backbone density for short helical segments, which were then combined with the heavy atom sites to further refine phases in PHENIX suite (*Adams et al., 2002*). Selenium substructures were obtained by running Phaser (*McCoy et al., 2007*) in its MR-SAD mode with phases from BASS-Hg poly-alanine model. These phases were subjected to automatic density modification with solvent flattening and histogram matching as implemented in PHENIX. Automatic chain tracing with the program RESOLVE yielded several helical fragments.

Manual tracing in the program COOT was used to fill the gaps (*Emsley and Cowtan, 2004*). For the native BASS, phase was obtained by rigid body refinement using SeMet BASS as the initial model. Model building was carried out using COOT. The selenomethionine and native BASS data were refined using PHENIX program to $R_{free}/R_{work}$ (0.27/0.24) and 0.25/0.22, respectively. The stereochemical quality of the models was assessed using the program PROCHECK and MolProbity. The native BASS model includes two chains with density observed for residues 640–791 and 660–795 for the two chains. The final selenomethionine BASS model, described here, includes two BASS subunits, including residues 640–802 and 648–802 of the two chains. Residues 803–843 were disordered and not observed in the BASS structures. All structural rendering figures were generated using UCSFchimera (*Pettersen et al., 2004*). The Kinesin-5 model was built using a combination of COOT, O and PyMOL programs (*DeLano, 2002*). We reasoned that the BASS domain is asymmetrically positioned in a phylogenetically conserved position within the Kinesin-5 primary sequence, 280 residues C-terminal to the motor domain and 120 residues N-terminal to the tail domain (*Figure 1*). Sequences between the BASS and motor domains (KLP61F residues 361–640) are likely helical with a high coiled-coil propensity, as measured by PAIRCOILS2, whereas sequences C-terminal to the BASS domain but upstream of the tail (residues 796–922) are predicted to form short helical segments, but with very low coiled-coil prediction scores (data not shown). To construct a model of the rigid Kinesin-5 central rod region, we overlaid a canonical parallel coiled-coil structure (tropomyosin; PDB ID 1C1G) onto the extreme N-termini of the BASS tetramer structure, where it has transitioned to a parallel coiled-coil conformation (C-alpha r.m.s.d. = 1.4 Å for 14 residues overlaid per chain). We extended each parallel coiled-coil region by 230 residues to form a 79-nm long rod.

## Major datasets

The coordinates and structure factor data for the Selenomethionine and native Kinesin-5 BASS structures were submitted to the Protein Data Bank (PDB) under the ID 4PXT and 4PXU, respectively.

## Acknowledgements

We thank Ruben Diaz-Avalos for electron microscopy data collection, John Voss for discussions on the Kinesin-5 BASS EPR spectroscopy data and Julie Leary, and Youjin Seo for help with mass-spectrometry. Kinesin-5 BASS diffraction data were collected at the Stanford Synchrotron Radiation Laboratory (SSRL), using 12-2 and 14-1 beamlines. We thank Peter Dunten, Ana Gonzalez, and Tzanko Doukov (SSRL) for help and advice with BASS data collection. JAB and SN thank Kevin D Corbett (Ludwig institute, UCSD) and Marijn Ford (University of Pittsburgh) for advice on BASS structure determination. JMS thanks Dr Andrew J Fisher for discussion leading to the inception of this project.

## Additional information

### Funding

| Funder | Grant reference number | Author |
| --- | --- | --- |
| National Institutes of Health | R00-GM08249, R01-GM55507, R01-GM05550712S1 | Jawdat Al-Bassam, Jonathan M Scholey |
| University of California Cancer Coordinating committee | | Jawdat Al-Bassam |

The funders had no role in study design, data collection and interpretation, or the decision to submit the work for publication.

### Author contributions

JES, Designed crystallization BASS constructs, Assembled, purified and crystallized BASS constructs, performed hydrodynamic analyses, Designed, purified and studied structure-based BASS mutants, Performed electron microscopy studies, Revised and suggested improvements to the manuscript., Conception and design, Acquisition of data, Analysis and interpretation of data; SN, Refined BASS crystallization, collected diffraction data and phase information, Determined and refined the BASS structures, Designed, purified and studied structure-based BASS mutants, Interpreted the BASS structure and prepared all the figures, Revised and suggested improvements to the manuscript., Conception and design, Acquisition of data, Analysis and interpretation of data, Drafting or revising

the article, Contributed unpublished essential data or reagents; JMS, Conceived the project, Interpreted the structure, Wrote, revised and suggested improvements to the manuscript., Conception and design, Analysis and interpretation of data, Drafting or revising the article; JA-B, Conceived the project, Designed crystallization BASS constructs, Determined and refined the BASS structures, Interpreted the structure and prepared all the figures, Wrote the paper, revised and suggested improvements to the manuscript, Conception and design, Acquisition of data, Analysis and interpretation of data, Drafting or revising the article, Contributed unpublished essential data or reagents

## Additional files

### Major datasets

The following datasets were generated:

| Author(s) | Year | Dataset title | Dataset ID and/or URL | Database, license, and accessibility information |
| --- | --- | --- | --- | --- |
| Scholey JE, Nithiananatham S, Scholey JM, Al-Bassam J | 2014 | BASS-Se structure | 4PXT; http://www.rcsb.org/pdb/explore/explore.do?structureId=4pxt | Publicly available at RCSB Protein Data Bank. (http://www.rcsb.org/pdb/home/home.do). |
| Scholey JE, Nithiananatham S, Scholey JM, Al-Bassam J | 2014 | BASS native structure | 4PXU; http://www.rcsb.org/pdb/explore/explore.do?structureId=4pxu | Publicly available at RCSB Protein Data Bank. (http://www.rcsb.org/pdb/home/home.do). |

The following previously published datasets were used:

| Author(s) | Year | Dataset title | Dataset ID and/or URL | Database, license, and accessibility information |
| --- | --- | --- | --- | --- |
| Whitby FG, Phillips Jr GN | 2000 | Crystal structure of tropomyosin at 7 angstroms resolution in the spermine-induced crystal form | 1C1G; http://www.rcsb.org/pdb/explore/explore.do?structureId=1c1g | Publicly available at RCSB Protein Data Bank. (http://www.rcsb.org/pdb/home/home.do). |
| Fujiwara Y, Minor DL | 2008 | Crystal Structure of an antiparallel coiled-coil tetramerization domain from TRPM7 channels | 3E7K; http://www.rcsb.org/pdb/explore/explore.do?structureId=3e7k | Publicly available at RCSB Protein Data Bank. (http://www.rcsb.org/pdb/home/home.do). |
| Yadav MK, Redman JE, Leman LJ, Alvarez-Gutierrez JM, Zhang Y, Stout CD, Ghadiri MR | 2005 | Structure based engineering of internal molecular surfaces of four helix bundles | 1UNX; http://www.rcsb.org/pdb/explore/explore.do?structureId=1unx | Publicly available at RCSB Protein Data Bank. (http://www.rcsb.org/pdb/home/home.do). |
| Ernst JA, Brunger AT | 2003 | High Resolution Structure of a Truncated Neuronal SNARE Complex | 1N7S; http://www.rcsb.org/pdb/explore/explore.do?structureId=1n7s | Publicly available at RCSB Protein Data Bank. (http://www.rcsb.org/pdb/home/home.do). |

**Reporting standards:** Standard used to collect data: We are following the Protein Data Bank standards to submit the coordinates and structure factor data.

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
