## [Decision Letter]

Thank you for sending your work entitled “Structural basis for the assembly of the mitotic motor Kinesin-5 into bipolar tetramers” for consideration at *eLife*. Your article has been favorably evaluated by a Senior editor, Tony Hyman as the Reviewing editor, and 3 reviewers.

The Reviewing editor and the reviewers discussed their comments before we reached this decision. All three reviewers were enthusiastic about your manuscript and in discussions we decided to accept it for publication as is. The reviewers had a number of comments that we think would improve the clarity of the manuscript, and these are included below. Please modify your manuscript appropriately, and upload a final version. You do not need to provide a cover letter saying what you did in this case.

1) Introduction: “Bipolarity is unique amongst MT motor proteins…” Not sure we know this. Crosslinking and sliding abilities have recently been described for Kinesin-8 and Kinesin-15.

2) Make the point that the bipolar K-5 tail structure may have to bear compressive loading as well as stretching?

3) Point out more explicitly that the heavily entwined 4-coil structure is expected to be torsionally stiff and that this can be key to making the rotational offset between the two ends impact the motor's preference to crosslink parallel or antiparallel MTs.

4) Results and Discussion subsection “Structure of BASS central bundle”: “Such a precise pattern... stabilizes its organization mechanically while transmitting forces...” Is it possible to roughly estimate how much stability in terms of, say, binding energy, the precisely alternating interfaces add to the overall tetrameric coiled-coil? Could it be possible that the total interaction energy might not be the most important feature of this novel construction principle, but maybe rather the possibility that this design could provide an efficient kinetic pathway to first form antiparallel dimers with the correct register before two of those associate to a tetramer?

5) Concluding remarks: “The 90**°** offset between the two dimeric ends..may set a larger rotational offset.” This is unclear. Larger than what? Isn't it the case, due to symmetry, that both dimeric catalytic motor head ends of the teramer rotate by the same amount if one were to fix the BASS domain in space, such that their relative angular orientation should remain the same?

6) Concluding remarks: “We hypothesize that this bundling is critical...” It is not so clear how or in what sense increased central rod stability would enable or improve force transmission. What kind of load on the coiled-coil stalk do the authors envision? I would think that the sketch in Figure 10 is highly schematic or do they envision lateral bending as important? Is there any evidence for that? A simple possibility would be that the stalk is always extensionally stressed. Compression seems unlikely because buckling would not need much force. Clearly a motor can push filaments apart in any direction while it is itself under extensional stress. In this vein, as far as the load on the motor goes, it does not make sense to distinguish between “inward pulling” and “outward pushing”. The difference between those two modes only affects the filaments that are being pushed or pulled, not the motors that could in both cases be either under compression or extension.

7) When modeling the C-termini of the motors, where does the tail end up with respect to the heads on that side? It seems, given the fact that the C-terminal stalk segments overhanging the BASS domain are much shorter than the N-terminal stalk fragments, that the tail would end up short of the heads.

8) It might be interesting to speculate, based on the new results, how the allosteric signaling from one end to other can take place. The way the two antiparallel coils are stapled together presumably prevents any relative sliding which could have been a possibility.

9) There appears to be some confusion in the figures and figure captions: Figure 2 caption somehow is mixed up a supplementary figure. Figure 3 caption is unclear: explain colors, give some more details. Figure 4 caption: “The lines in panel C” should read “The lines in panel B”? Figure 5 caption also seems to have mistakes and inconsistent formatting.

10) In Figure 7 there seem to be skip residues in some positions. This is hard to make out because the asterisks in some places bridge two positions – please adjust and mark positions where there is an octad rather than a heptad pattern. Please explain in the legend of Figure 7 what the asterisks indicate.

11) This is the first and only crystal structure of a kinesin tail to date and this is worth stating.

12) Do the temperature factors provide any clues to dynamics?

13) “Interface” is overused – this same term is used for the helical seams between neighbouring chains and for the axial boundaries between hydrophobically and electroststically-stabilised sections of tetracoil.

14) Given that *eLife* likes to buck convention, perhaps the authors could consider contributing their PyMOL .py session files so that readers can have a 3D look at colour-coded structures?

15) The hierarchy of interactions stabilising the 4-coil could be better described. The diamond profile beyond the elbows makes it clear that the tetracoil in these regions is the interaction of two coiled coils, but elsewhere the spacing between the chains is much more similar and it looks much more as though each chain makes quasi–equivalent interactions with 3 others? Perhaps make this explicit, or else make it clear why this is not so.

[Editors’ note: there is not an accompanying Author response to these minor comments.]